# Polyamine Metabolism under Different Light Regimes in Wheat

**DOI:** 10.3390/ijms222111717

**Published:** 2021-10-29

**Authors:** Orsolya Kinga Gondor, Judit Tajti, Kamirán Áron Hamow, Imre Majláth, Gabriella Szalai, Tibor Janda, Magda Pál

**Affiliations:** Eötvös Loránd Research Network, Centre for Agricultural Research, 2462 Martonvásár, Hungary; gondor.kinga@atk.hu (O.K.G.); tajti.judit@atk.hu (J.T.); hamow.kamiran@atk.hu (K.Á.H.); majlath.imre@atk.hu (I.M.); szalai.gabriella@atk.hu (G.S.); janda.tibor@atk.hu (T.J.)

**Keywords:** light intensity, plant hormones, polyamines, putrescine, spermidine, spermine, wheat

## Abstract

Although the relationship between polyamines and photosynthesis has been investigated at several levels, the main aim of this experiment was to test light-intensity-dependent influence of polyamine metabolism with or without exogenous polyamines. First, the effect of the duration of the daily illumination, then the effects of different light intensities (50, 250, and 500 μmol m^–2^ s^–1^) on the polyamine metabolism at metabolite and gene expression levels were investigated. In the second experiment, polyamine treatments, namely putrescine, spermidine and spermine, were also applied. The different light quantities induced different changes in the polyamine metabolism. In the leaves, light distinctly induced the putrescine level and reduced the 1,3-diaminopropane content. Leaves and roots responded differently to the polyamine treatments. Polyamines improved photosynthesis under lower light conditions. Exogenous polyamine treatments influenced the polyamine metabolism differently under individual light regimes. The fine-tuning of the synthesis, back-conversion and terminal catabolism could be responsible for the observed different polyamine metabolism-modulating strategies, leading to successful adaptation to different light conditions.

## 1. Introduction

Polyamines (PAs) are low-molecular-weight, aliphatic biogenic amines with variable number of amino groups and are ubiquitously distributed in all cells [1]. Due to their various interactions with macromolecules, such as nucleic acids, phospholipids, proteins, pigment-protein complexes or with phenolic compounds, PAs are involved in the regulation of numerous cellular, physiological, and biochemical processes [2]. In addition, it is supported by more and more evidence that PAs exert crucial roles in plant signaling during abiotic and biotic stress responses [3,4]. Not surprisingly, this means that the fine-tuning of the PA metabolism, which is linked to the synthesis of other protective compounds and signaling molecules and results in an optimum PA content and ratio, is very important. The PA metabolism is dynamic due to the synthesis and catabolism, completed with the back-conversion of higher PAs observed in the PA-cycle; furthermore, the role of conjugation and transport is also evident, which has been well documented in several recent reviews [2,5,6].

The key role of PAs in plant development has been proven by the demonstration of correlation between plant production parameters and endogenous PA contents [7]. On the other hand, exogenous PA treatments have also been reported to have positive effects under optimal growth conditions or to provide protection against various stress factors in different plant species [8,9]. However, the beneficial effects do not only depend on the applied PA compounds, the concentrations and plant species, but also on the plant genotype [10,11]. It should also be taken into consideration that the observed differences were always linked to the alteration or shift in the PA metabolism due to its overlap with the synthesis of proline or phytochelatins and/or the different intensity of the steps of the PA-cycle [12,13]. In several cases, relationships between changes in plant hormone contents, such as salicylic acid (SA) or abscisic acid (ABA) and PAs, have been reported under different conditions, suggesting that PAs can influence the hormonal balance, which was also found to be responsible for the modulation of the putative protective effect of PAs [10,14,15].

The connection between PAs and photosynthesis was recognized a long time ago, based on the evidence of high concentration of PAs in the chloroplast thylakoid membranes, the light harvesting complexes and PS II complexes, even in the PS II reaction centers themselves [16,17] and the high activity of the transglutaminase enzyme (TGase), which catalyzes the covalent binding of PAs to pigment-proteins [18,19]. Light controls PA biosynthesis [20]. The changes in the activity of both PA synthesis enzymes—arginine decarboxylase (ADC), ornithine decarboxylase (ODC), spermidine synthase (SPDS), spermine synthase [21,22,23]—and catabolite enzymes—polyamine oxidase (PAO) [22,24,25], or TGase—have been reported in the presence of different light conditions [24,26]. It has also been proven that PAs are able to influence photosynthesis at several levels: for example, they can modify the chlorophyll destruction and/or biosynthesis; binding to photosynthetic complexes can induce conformal changes; as buffer compounds in the thylakoid lumen, they may increase chemiosmotic ATP synthesis [27].

Despite the above-described relation between PA metabolism and light, there is still little direct information available about how light quantity influences the PA metabolism (PA contents and/or activities or gene expression levels of PA synthesis enzymes). For the first time, in this study, research was carried out in order to reveal how light regime influences the effects of exogenous PAs on PA metabolism in wheat plants. According to these, the main aim of the present study was to highlight the effects of illumination on PA metabolism. In order to achieve our goal, firstly, the effect of light period was tested, after which different light intensities were applied. As one of the potential ways to stimulate PA metabolism is the application of exogenous PAs, three different light intensities were used in combination with PA treatments. The following questions should be considered: (1) How do light quantity and shorter or longer day/night periods with the same light intensity modify the PA pool, and how do different light intensities with the same light/dark period influence PA metabolism? (2) How do light conditions influence the putative positive effect of PAs with special regard to which part of the PA metabolism is the most sensitive for light and vice versa? (3) How can PA treatments reduce the effect of lower light condition on the level of PA metabolism? An additional aim was to reveal the effect of light in combination with PA treatments on the changes in SA and ABA syntheses.

## 2. Results

### 2.1. Effect of Hours of Light Per Day

#### 2.1.1. Effect of Different Duration of Illumination on Pigment Contents

In the first part of the experiments, the main task was to monitor the effect of different durations of illumination; therefore, a shorter (8 h/16 h light/dark) and a longer daily light period (16 h/8 h light/dark) was applied. The quantities of photosynthesis-related plant pigments, namely chlorophyll a and b, and typical, abundant carotenoids (including trans-β-carotene, 9-cis-β-carotene, trans-violaxanthin, trans-neoxanthin, trans-lutein) were determined. Sampling was performed at the light period of 8/16 h light/dark condition starting at the beginning of illumination (0 h, 1 h, 2 h, 3 h, 5 h and 7 h), in comparison with the early lighting period of 16 h/8 h light/dark period (0 h, 1 h, 2 h, 3 h, 5 h, 7 h and 9 h). The sampling at 0 h means that the plants were sampled in the dark, directly before the onset of the light.

The investigated pigment contents were higher in the plants grown under longer daily illumination period, except for pheophytin a (Table 1). The most pronounced difference was observed in the case of chlorophyll a, where the initial chlorophyll a content in the plants grown under lower light/dark period was only 76% of that measured under 16 h/8 h day/night period. Under 8 h/16 h day/night light condition, most of the pigment content showed increasing tendency and showed accumulation already after the first hour, except for pheophytin a and 9-cis-violaxanthin, which decreased during the investigated period. Although in several cases, significant differences were observed under both light conditions, the 16 h/8 h day/night condition caused more slight fluctuations in the pigment content during the 9 h period started at the beginning of illumination (Table 1).The changes in the levels of the three most abundant pigments (chlorophyll a, chlorophyll b, and trans-lutein) showed opposite tendency under the two light conditions, as they increased under 8 h/16 h, but decreased under 16 h/8 h light/dark conditions, resulting also in a different Chla/Chlb ratio with ranges of 1.99–2.04 and 2.11–2.16, respectively.

#### 2.1.2. Changes in Polyamine Patterns under Different Durations of Illumination

Difference in the light period resulted in pronounced alteration in the PA pattern in both the leaves and roots (Figure 1). Although SPD was the most dominant PA in the leaves and roots of plants grown under 16 h/8 h day/night condition (Figure 1B,F), fewer hours of light shifted the proportion in the direction of 1,3-diaminopropane (DAP), the terminal catabolite product of higher PAs both in the leaves and roots (Figure 1D,H).

It was also found that during the day, the PA contents changed significantly in the cases of PUT, SPM and DAP. The PUT level in the leaves increased under both light conditions from the start of the illumination, it was especially pronounced under 8 h/16 h treatment (Figure 1A). In the roots, the changes in PUT content showed decreasing tendency under 16 h/8 h day/night condition and did not change significantly at 8 h/16 h condition (Figure 1E). Although under these conditions the light parameter did not influence the SPD content (Figure 1B,F), the SPM level increased under 16 h/8 h light condition from the start of the illumination both in the leaves (Figure 1C) and roots (Figure 1G) and also under 8 h/16 h treatment, but only in the leaves (Figure 1C).

These results revealed that although the ratio of PAs in the leaves was different under the two hours of light/day conditions, the changes in the PUT/(SPD + SPM) ratio showed similar patterns during the day, starting from the onset of illumination, as it increased under both conditions. However, in the roots, this ratio decreased only under longer illumination per day (Figure 2).

### 2.2. Effect of Light Intensity in Combination with Different Polyamine Treatments

Based on the results of the first experiment, in the second experiment the effects of light intensity were tested in plants grown under 16 h/8 h light/dark period, and leaf and root samples were collected after 4 h of illumination. Plants were grown either at elevated (L1: 500 µmol m^−2^ s^−1^), medium (L2: 250 µmol m^−2^ s^−1^) or low (L3: 50 µmol m^−2^ s^−1^) light intensities.

#### 2.2.1. Changes in Fluorescence Induction Parameters and Pigment Contents

In order to monitor the physiological status and photosynthetic activity of plants, chlorophyll a fluorescence measurements were carried out. The maximal quantum yield of photosystem II (Fv/Fm parameter) was not influenced during the experiment either by the growth light or by PA treatments, showing an average value of 0.792 (Appendix A).

Under control conditions, without any PA treatment, the actual quantum yield of PSII [Y(II)] parameter was higher at L1 than at L2 or L3, while all PA treatments reduced the differences in it between L1 and L2 or L3 by increasing the value of Y(II) mainly in L2 and L3 (Figure 3). Interestingly, under L1 only SPM treatment could significantly increase the photochemical yield. At L2 and L3 all PA treatments had statistically significant positive effects on Y(II) compared to the relevant controls (without PA treatments). Corresponding tendency was observed in other fluorescence induction parameters, such as quantum yield of regulated way of energy dissipation [Y(NPQ)], quantum yield of unregulated means of energy dissipation [Y(NO)] parameters and the linear electron transport rate (ETR) presented in the Appendix A.

Low light intensity provoked a decrease in the level of all the presented plant pigments (Table 2). The most pronounced decrements were observed in the case of trans-violaxanthin, chlorophyll a and trans-β-carotene with 48, 29 and 37% decrease under L3 light condition compared to the values at L1, respectively. Also, a decreasing tendency of Chla/Chlb ratio was observed under L1, L2 or L3 with values of 2.33, 2.28 and 2.18, respectively. PA treatments, especially PUT, mainly also decreased the plant pigment concentrations under L1 light condition. Under L2 light conditions, PA treatments did not cause significant differences compared to the relevant control, while slight, not always statistically significant but clearly increasing effect was detected at L3 after PUT and SPD treatments in the case of trans-lutein, chlorophyll a and b and 9-cis-β-carotene levels.

#### 2.2.2. Differences in the Polyamine Metabolism

Different light intensities caused pronounced differences in the PA contents in the leaves but not in the roots. In the leaves, in parallel with the decrease of the light intensity, the level of PUT and SPD decreased, the amount of SPM did not change, while the DAP content increased. In the roots, no remarkable alteration could be detected in the PA contents of plants without any PA treatment under the different light intensity conditions.

In general, the PA treatments increased the endogenous PA contents both in the leaves and roots, interestingly especially under L2 condition in the leaves and under L3 condition in the roots. 

Under the highest light condition, exogenous PAs did not influence the endogenous PA content in the leaves, but under lower light intensities (L2 and L3) the initially lower PUT contents showed increments (Figure 4A). Although the initial SPD content was also lower under L2 than it was found under L1, SPD and SPM treatments could double its level, while under L3 the further decreased initial SPD level could not be influenced by the PA treatments (Figure 4B). None of the treatments could influence the SPM level (Figure 4C). The leaf DAP content was slightly increased after PUT application under L1 and after SPM treatment under L2, but as its level was already high under L3 light intensity without PA application, exogenous PAs could hardly affect it (Figure 4D). In parallel with these, in the roots the changes of PUT, SPM and DAP contents showed almost the same pattern (Figure 5A–D). Namely, the light conditions alone did not influence them, but the PA treatments induced their accumulation especially under the lowest light intensity (L3). The SPD content showed only slight and similar but statistically not significant increase after PA treatments under different light conditions (Figure 5B).

Overall, a positive correlation was found between the light intensity and total PA contents of the leaves. In addition, it appears that the highest accumulation inducing effect of the PA treatments could be detected under L2 in the leaves, while in the roots it could be found under L3 conditions, compared to the relevant control. Nevertheless, the highest PUT/(SPD + SPM) ratio was found after PA treatments under L3 conditions in both the leaves and roots (Figure 6).

The two main pathways responsible for PUT synthesis are the arginine and the ornithine pathways, catalyzed by arginine decarboxylase (ADC) and ornithine decarboxylase (ODC), respectively. The light intensity alone did not influence the ADC or ODC expression. Despite the pronounced PA treatment-induced increases of leaf PUT levels, especially in the case of higher PAs under L2 or L3 light conditions, the gene expression level of *ADC* in the leaves did not change remarkably, except for a slight but statistically significant decrease of it in PA-treated plants under L2 (Figure 7A). At the same time, the transcript level of *ODC* showed a slight, not always statistically significant, increase after SPM treatment under all light conditions (Figure 7B). From PUT the higher PAs (SPD and SPM) are synthesized with the involvement of spermidine synthase (SPDS) and S-adenosylmethionine decarboxylase (SAMDC) and their gene expressions were not modified by the light quantity. However, the gene expression pattern of *SPDS* and *SAMDC* in the leaves showed slight decreasing tendencies after PA treatments under all light conditions, with the lowest expression found after SPM application (Figure 7C,D). Nevertheless, the initial *SPDS* transcript levels and changes in them did not correlate with the differences in the SPD or SPM contents measured after the different treatments. The expression of *perPAO*, which encodes the peroxisomal PAO responsible for the back-conversion of higher PAs to PUT, was only induced under L1 light after PA treatments in the leaves; but at lower light intensities, it was inhibited by the exogenous PAs (Figure 7E). At the same time, the expression level of polyamine oxidase (PAO), which is responsible for the terminal catabolism of PAs, was induced under L3 conditions after all PA treatments, especially SPM (Figure 7F).

In contrast to the leaves, in the roots the PA-induced level of *ADC* transcript under L2 and L3 lights and that of *ODC* under L1 and L2 was in parallel with the higher PUT levels, which means that exogenous PAs induced in vivo PUT synthesis (Figure 8A,B). In the case of *ODC*, the expression level showed a dramatic increase after PUT treatment at L1 and L2 (Figure 8B). However, both *ADC* and *ODC* expressions were inhibited by PUT treatment under the lowest light condition, possibly due to the negative feedback mechanism induced by the taken up PUT. The gene expression pattern of *SPDS* in the roots was similar to the one observed for the leaves, and although especially the SPM treatment increased the endogenous SPD and SPM contents under all the applied light conditions, the *SPDS* expression could not have been responsible for these alterations (Figure 8C). The same can be claimed for SAMDC (Figure 8D). In addition, none of the genes involved in PA synthesis (*ADC*, *ODC*, *SPDS* and *SAMDC*) were influenced by the light intensity alone. Like in the case of the leaves, the highest level of DAP was observed in SPD- and SPM-treated plants under L3 light, where lower *perPAO* transcript levels were measured in the roots (Figure 8E). The gene encoding the investigated polyamine oxidase (PAO) isoform was not expressed in the roots.

In relation to the terminal catabolism of PAs, the contents of two important compounds—β-alanine and γ-aminobutyric acid (GABA), which are both involved in stress responses—were determined. β-alanine is synthetized from DAP, while GABA is synthesized during the terminal catabolism of PUT via Δ^1^-pyrroline in the reactions both catalyzed by NAD^+^-dependent 4-aminoaldehyde dehydrogenase (AMADH). Thus, changes in the levels of these compounds may also reflect the intensity of the catabolite side of PA metabolism. Despite the observed differences in the leaf DAP content, the amount of β-alanine in the leaves did not change remarkably during none of the treatments (Figure 9A). Interestingly, although DAP content in the roots increased by exogenous PAs especially at L3, the level of β-alanine increased after PA treatments only under L1 and L3 light conditions with the highest level in the SPM-treated plants at L1 (Figure 9B). The changes in GABA content also showed partly different patterns compared to its precursor, PUT. Although in the leaves GABA level decreased under the lowest light intensity as it was found for leaf PUT content, its levels decreased after PA treatments, especially under L1 and L2 light condition (Figure 9C) in contrast to the amount of PUT. In the roots GABA content also decreased under L3 compared to L1 like the level of PUT, but increased after SPM treatment in plants grown only at L1 and L2 (Figure 9D).

#### 2.2.3. Differences in the Salicylic Acid and Abscisic Acid Contents and Synthesis

As previous results revealed that PA treatments influence the synthesis of certain plant hormones, one of our special aims was to reveal the relationship between the PAs metabolism and SA and ABA contents and synthesis, under different light intensity conditions. At lower light intensities, lower SA level was detected, especially in the roots (Figure 10A,C). Among PAs, only SPM treatment could provoke significant changes, as the level of leaf SA increased under L2 but decreased under L1 and L3 (Figure 10A). In contrast, the ABA content was lower at L1 and PUT and SPD treatments increased it, while SPM decreased it (Figure 10B). Slight increase in the ABA content was observed after the PA treatments, under L2 and L3 conditions (Figure 10B).

In the roots, besides the positive correlation between the light intensity and the SA content under control condition, the PA applications caused further changes in the SA level depending on the light intensity, as exogenous PAs under L1 decreased, under L2 they increased it, while under L3 only SPD could increase it slightly (Figure 10C). In contrast, the initial ABA content was similar under all light conditions, while exogenous PA influenced it in a similar way, namely it increased it in all cases (Figure 10D).

The gene expression levels of the key enzymes involved in SA and ABA synthesis were also determined under the different light conditions and PA treatments. In the leaves, the phenylalanine-ammonia-lyase (*PAL*) expression level did not show relation to the changes in the leaf SA content, as under L1 and L2 conditions slight inhibition, but under L3 pronounced induction of it was found after the treatments with SPD or SPM (Figure 11A). The chorismate synthase (*CS*) transcript level was hardly influenced by the applied treatments (Figure 11B), while the expression level of isochorismate synthase (*ICS*) was significantly increased by PUT treatment under L1 and L2 light conditions and decreased under L3 after SPD or SPM treatments (Figure 11C). Thus, the only direct connection between the gene expression data and SA content in the leaves is that the lowest SA content and *ICS* expression was measured in the SPM-treated plants under L3 light conditions. The changes in 9-cis-epoxycarotenoid di-oxygenase (*NCED*) expression show only a few statistically significant differences, as under L1 its transcript level decreased in all the PA treatments, and under L3 light conditions it decreased in SPM treatment (Figure 11D).

Interestingly, in the roots, the expression of *PAL* showed similar pattern to the changes of SA content under L1 conditions (Figure 12A), while under L2 rather the expression levels of *CS* and *ICS* showed correlation with the changes of the root SA content (Figure 12B,C). Under L3 growth conditions, no reasonable changes were observed in the gene expression levels of *PAL*, *CS* or *ICS*. The PA treatment-induced changes in the expression level of *NCED* compared to the relevant control in the roots showed correlation with the root ABA content only under L1 conditions, as both the ABA and the *NCED* transcript levels increased after all PA treatments (Figure 12D).

## 3. Discussion

Numerous valuable studies have been published on the investigation of PA content and metabolism either under normal growth conditions or during biotic or abiotic stresses in various plant species [28,29,30,31,32,33,34,35,36,37,38]. On the other hand, several studies have been published on monitoring the protective effect of exogenous PAs [8,9,10,11,13,14,39,40,41]. Exogenous PAs provided protection under osmotic and cadmium stress [8,9], but the mode of action and the degree of protection depended on the alteration in the PA metabolism; thus, it has been demonstrated that “the more PA, the better” statement cannot be generalized [10,11,13]. Most of these experiments were carried out in growth chambers with photosynthetic photon flux density (PPFD) in the range of 100–270 µmol m^2^ s^−1^ with one day/night regime [8,9,10,11,13,14,22,29,32,34,36]; some of them were conducted under higher light conditions [30,31,41] and none of them under different hours of illumination. In addition, only a few studies are available on the comparison of changes in dedicated components of the PA metabolism under different light conditions [21,22,23,24,25,26], and none of them reported the simultaneous effect of exogenous PA treatments. Therefore, in the present study, our aim was the better understanding of the effect of light on PA metabolism and how the exogenous PA treatments may modify the interaction of PAs and light.

### 3.1. Relationship between Light Quantity and PA Metabolism

#### 3.1.1. Modulating Effect of Light on PA Metabolism

Adaptive changes in plant pigment content to different light conditions have been well-studied in various plant species [42,43]. In the present study, the daily lighting regime (8 h/16 h or 16 h/8 h light/dark periods) remarkably influenced the pigment pattern from the start of illumination in the leaves of wheat plant, indicating that light regime and the hours of illumination induced sufficient changes in the plant metabolism to investigate the PA pool.

The most detailed study on the effect of lighting hours on PA metabolism was carried out in tomato. Changes in PUT, SPD and SPM levels were measured in the leaves of tomato plants grown under 12 h/12 h light/dark period, after which the plants were exposed to 24 h continuous light or dark conditions. It was found that the PA content, especially that of PUT and SPM increased under light conditions and reached maximum values at 6–12 h, but after that it decreased to the initial levels [22]. In the present study, under the two different light periods (8 h/16 h or 16 h/8 h light dark), the changes in the PA pattern showed some similarities. Despite the evidence of the altered ratio of the individual PA compounds, the PUT and SPM content in the leaves increased, but leaf and root SPD levels did not change from the beginning of the illumination under both regimes. The longer lighting hour condition resulted in pronounced decreasing tendency of leaf and root DAP and root PUT, but increment of root SPM was detected. The results also showed that the leaves and roots responded partly differently, with higher impact of light on the leaves, manifested in the increase of the total PA concentration and with the decrease of the level of DAP, suggesting the inhibition of the terminal catabolism of higher PAs under light condition (Figure 13).

#### 3.1.2. Influence of PA Treatments on the Effect of Light on PA Metabolism

Based on the result of the first experiment, in a second step, we investigated the effect of light intensity under 16 h/8 h light/dark condition. The light quantity under three different light intensities (50, 250 or 500 μmol m^−2^ s^−1^) showed close correlation with the plant pigment contents. However, PA treatments could hardly influence the pigment contents significantly compared to the relevant controls, except for PUT. The measurement of chlorophyll a fluorescence induction parameters was performed in order to gain information about the physiological status of the plants. The decrease in light intensity induced pronounced changes in the chlorophyll a induction parameters, but interestingly especially higher PAs (SPD and SPM) could reverse them, even under the lowest L3 intensity. PUT, SPD and SPM treatments have also been reported to increase the Fv/Fm and photochemical quenching chlorophyll fluorescence parameters and photosynthetic pigment concentration under salt stress conditions in Bakraii citrus [44]. In addition, PA treatments have shown to have protective effect against osmotic stresses in various plant species, which was related to the stimulation of photosynthesis, suggesting that PAs may protect the photochemical capacity [11,45,46,47].

The observed PA-induced photosynthetic improvement manifested as an increase in the quantum yield of PSII in SPD- and SPM-treated plants, making it important to reveal the possible role of and the changes in the PA metabolism. In the second experiment the observed higher DAP content in the leaves but lower PUT and SPD under lower light intensity showed similar pattern to the results of the first experiment during lower illumination period (Figure 13). Although the light intensity alone did not influence the gene expression levels of enzymes involved in the PA synthesis or catabolism, the differences in the total PA contents in the plants grown under different light regimes in both of the experiments confirm the role of light in PA accumulation [22,23,27].

Appendix A represents the significant changes in the PA metabolism-related changes regarding the determined contents and gene expression levels compared to the relevant controls. Light alone influenced PA metabolism in the leaves but did not affect it in the roots. However, the effects of PA treatments manifested differently due to the diverse light intensities in both the leaves and roots (Figure 14A,B).

The most remarkable changes in the leaf PA pool were detected under L2 condition. Like in the case of these results, earlier in wheat and maize at 250 µmol m^−2^ s^−1^, the PA treatments increased the endogenous PA content and not only the content of the applied and taken up PA increased, but due to the PA-cycle, other PA compounds also accumulated [9]. According to our results, under the highest and lowest light intensities the effect of the light was dominant, while under L2 light condition the influence of the PA treatments was more pronounced on the endogenous PA pool. Under L2 light conditions, the higher PA treatments induced similar total PA levels in the leaves as it was detected under L1 with or without any PA, which was due to the higher SPD accumulation. However, under L3 none of the PA treatments could compensate the decrease of total PA content resulting from the lower light intensity. As under L1 the initial SPD level was already higher, PA treatments induced *perPAO* expression, while under L2 and L3 conditions, where the initial SPD content was already lower, the PA treatments decreased its expression. As exogenous PAs increased PUT and SPD levels under L2, as a feedback inhibition, a decrease in *ADC*, *SPDS* and *SAMDC* gene expression levels were observed. Under the lowest light condition, the PUT content increased after PA treatments but the already low SPD contents could not be decreased further by the PA treatments. In addition, the highest DAP accumulation was also detected here either with or without exogenous PAs, suggesting that the terminal catabolism of higher PAs was determining at L3. This was also supported by the activation of *PAO* expression in these leaf samples (Figure 14A).

Compared to the results in leaves, in the roots the highest total PA amount was detected in the SPM-treated plants under all light conditions, but especially under L3, due to the higher PUT and SPM levels. However, DAP content was also the highest in SPD- and SPM-treated plants grown under L3. PA treatments induced in vivo PUT synthesis, especially under L2 condition, like in the case of our previous results in wheat [8,12]. As a feedback response, the increased levels of root SPD and SPM after PA treatments under L3 conditions could cause a decrease in the level of *SPDS* transcript. According to these results, in the roots, PA treatments had the greatest effect on endogenous PA contents under the lowest light intensity, where both the back-conversion and the terminal catabolism, but also the in vivo synthesis could be responsible for the increased amounts of PUT and in turn that of DAP. At the same time, the levels of β-alanine and GABA were the highest in the SPM-treated plants under L1 condition, suggesting the more efficient canalization of the metabolism.

Apart from the several functions of β-alanine and GABA as endogenous signaling molecules in plant growth regulation and plant development, these compounds are also involved in multiple stress responses [48,49]. However, changes in the levels of these compounds do not relate closely to the changes in PA content, and in certain cases the effect of light modulate their production differently compared to the PA metabolism itself.

### 3.2. Effect of Light Quantity on Salicylic Acid and Abscisic Acid Synthesis, and its Relation to PA Metabolism

Light is also required for SA biosynthesis and SA perception [50,51,52]. Limited and contradictory literature are available on the effect of light on SA synthesis [53,54,55]. However, investigation on Arabidopsis mutant plants with constitutively high SA levels and on the effect of exogenous SA showed that controlled level of SA is necessary for optimal photosynthesis [56,57]. Under the present conditions, in the second experiment, there was only slight positive correlation between the light intensity and SA content in the leaves, but it was pronounced in the roots of the wheat plants. However, these differences did not manifest in the expression levels of the genes encoding key enzymes of SA synthesis.

Close positive correlation between SA and PA contents was found in several cases under various conditions [58,59,60,61,62]. Mutual influence of the PAs and SA on each other’s synthesis have already been recognized [8,22,63,64,65,66]. The relationship between SA and PAs was further proven by the finding that 0.5 mM SPD and SPM in hydroponic solution increased the SA content in both the leaves and roots of wheat and maize [9]. Under the present conditions, in the leaves, only SPM treatment could influence the SA content, as under L1 and L3 it decreased, but under L2 it increased. Although the expression level of *PAL* dramatically increased in SPM-treated plants under L3, this could be related only to the effect of higher PAs on the synthesis of phenolic compounds, which are more dominant under low light condition [39,67]. Changes in the expression pattern of *CS* or *ICS* were not closely related to the SA content, except for the SPM-treated plants under L3, where the lowest SA and *ICS* expression was measured. In contrast, in the roots, increase in the level of *ICS* transcript was in accordance with the increase of SA content under L2 condition after PA treatments. It should also be mentioned again that the effect of PA treatments on SA were the most dominant under L2 light conditions, where similar light conditions were applied and results were obtained like previously in wheat [9].

Light has also been suggested to regulate ABA biosynthesis directly or indirectly [68], however, the literature data on the light-induced changes in ABA content are contradictory [69,70,71,72]. Earlier findings suggested a positive feedback loop between ABA and PAs [61,73]. For example, overexpression of the genes of PA synthesis enzymes, such as *ADC*, *SAMDC* or *SPMS*, also resulted in increased ABA biosynthesis due to the higher expression level of *NCED* [62]. In the present experiment the decrease of light intensity alone only slightly increased the leaf ABA content but did not influence it in the roots. Also, only slight changes were observed after the PAs treatments in the ABA level of the leaves. In contrast, in the roots, all the applied PA treatments increased it: under higher light condition, especially PUT and SPD, while under lower light condition SPM treatment had the greatest effect. However, the *NCED* expression level was only in correlation with the root ABA content under the highest light condition. Like in the case of these results, PA treatments also increased the ABA level in Arabidopsis but only in the leaves [14]. PA treatment also increased ABA content especially in the roots under similar light condition like L2 in the present experiment in wheat [11,12].

Correlation analysis revealed that in the leaves significant positive relationship exists only between PUT and SPD, and between SPD and SA content (Table 3). In the roots, the amounts of PUT showed strong, positive correlation with SPD and DAP levels, while SPD showed the same correlation with ABA content (Table 4).

## 4. Materials and Methods

### 4.1. Plant Materials, Growth Conditions and Treatments

In the present study, winter wheat (*Triticum aestivum* L.) variety ‘Mv Béres’ was used. Germination was carried out on moistened filter papers for 3 days at 26 °C in the dark, after which wheat seedlings were transferred to modified Hoagland solution (15/plastic pot) [74]. The plant growth solution was changed every two days. 

In the first experiment, where the effect of the duration of light on the PA metabolism was investigated, Conviron PGR-15 plant growth chamber (Controlled Environments Ltd., Winnipeg, MB, Canada) equipped with high intensity discharge (HID) metal-halide lamps was used and plants were grown under the following conditions: 22 °C/20 °C day/night temperature, 75% relative humidity either with 16 h/8 h light/dark or 8 h/16 h light/dark periodicity. The photosynthetic photon flux density (PPFD) was 190 μmol m^−2^ s^−1^ under both conditions. 14-day-old plants were sampled for further analyses, before the start of illumination, under dark (0 h), and 1, 2, 3, 5, 7 and 9 h after the beginning of the light condition. In the case of the 8 h/16 h light/dark period, only 6 sampling times were applied.

In the second experiment, plants were grown under different light intensity conditions: at 500 (elevated light: L1), 250 (medium light: L2) and 50 (low light: L3) µmol m^2^ s^−1^ LED light [75], at 22 °C/20 °C with a photoperiod of 16 h/8 h (day/night condition) and 75% relative humidity in Conviron PGR-15 plant growth chambers (Controlled Environments Ltd., Winnipeg, MB, Canada). The 7-day-old plants were treated with nutrition solution containing 0.3 mM PUT, SPD or SPM (this concentration and the duration of the treatments were chosen based on our previous results [9]). After a 1-week exposure to different PA treatments, fully developed leaves and roots were collected. The sampling started after 4 h of illumination, as this time interval is sufficient for PA contents to reach medium level during the day.

### 4.2. Chlorophyll a Fluorescence Induction (FI) Analysis

The FI analysis was carried out using pulse-amplitude-modulated fluorometer (PAM) with a blue LED-Array Illumination Unit IMAG-MAX/L (λ = 450 nm) (Imaging-PAM MSeries, Walz, Effeltrich, Germany). Leaves were exposed to dark for 15 min in order to reach the open state of the acceptor side of the photosynthetic machinery. Afterwards, the maximum quantum yield of PSII (Fv/Fm) parameter [76] was determined using a saturation light for 0.8 s duration and with 3000 μmol m^–2^ s^–1^ intensity.

The quenching analysis of chlorophyll-*a* fluorescence was carried out using a 270 μmol m^–2^ s^–1^ actinic light intensity until the steady-state level of photosynthesis (duration: 15 min). During this period, the saturation light defined above was applied with 30 s frequency provided by the IMAG-MAX/L unit.

The actual quantum yield of PSII [Y(II)], the quantum yield of regulated way of energy dissipation [Y(NPQ)] and quantum yield of non-regulated way of energy dissipation [Y(NO)] parameters [77] and the linear electron transport rate (ETR) were calculated during the entire analysis.

### 4.3. Polyamine Analysis

After homogenizing 200 mg samples with 2 mL of 0.2 N perchloric acid, the extract was centrifuged at 10,000× *g* at 4 °C for 10 min and the supernatant was used for the pre-column derivatization with dansyl chloride (Merck-Sigma group, Darmstadt, Germany) [65]. PUT, SPD and spermine (SPM) were analyzed together with DAP, the terminal catabolic product of higher PAs (SPD and SPM), on a reverse phase Kinetex column (C18, 100 × 2.1 mm, 5 μm, Phenomenex, Inc., Torrance, CA, USA) by HPLC, using a W2690 separation module and a W474 scanning fluorescence detector with excitation at 340 nm and emission at 515 nm (Waters, Milford, MA, USA).

### 4.4. Extraction of Plant Hormones and Analytical Procedure

Extraction, separation and detection with tandem mass spectrometry (UPLC-MS/MS) were carried out according to [78] with slight modifications, according to [11]. Leaf or root samples were ground in liquid N_2_ and extracted with methanol:water (2:1) to a final sample ratio of 100 mg FW ml^−1^. Solvents used were all at least HPLC grade and were purchased from VWR International (Radnor, Pennsylvania, United States). UPLC-MS/MS analysis was performed on a Waters Acquity I class UPLC system coupled to a Waters Xevo TQ-XS (Milford, MA, USA), equipped with a UniSpray ion source (US) operated in timed MRM mode, with argon (Gruppo SIAD, Bergamo, Italy) as a collision gas. Separation was performed on a Waters Acquity HSS T3 column (1.8 μm, 100 mm × 2.1 mm), at 40 °C. For gradient elution, water and acetonitrile containing 0.1 *v/v* % formic acid were used. Data processing was performed using Waters MassLynx 4.2 and TargetLynx software (Milford, MA, USA).

### 4.5. Pigment Extraction and Analyses

Liquid N_2_-homogenized 200 mg fresh weight leaf plant tissue was measured into non-transparent 2 mL safety Eppendorf tubes and spiked with beta-apo-8′-carotenal (Merck-Sigma group, Darmstadt, Germany) as an internal standard at 2.5 or 5 µg 100 mg^−1^. Samples were extracted twice, each round with 1 mL of acetone:methanol 80:20 *v/v*%, by vortexing for 10 s, followed by shaking in a MiniG 1600 instrument (SPEX SamplePrep.; Metuchen, NJ, USA) in a cryo-cooled rack at 1250 rpm for 3 min. After centrifugation at 14,000× *g* (at 4 °C for 10 min), supernatants were collected, pooled and filtered through 0.22 µm PTFE syringe filters and analyzed immediately. For LC-PDA-MS analysis a Waters Acquity I-class UPLC coupled to Xevo TQ-XS Mass spectrometric system and a Thermo Accucore C30 2.6 µm, 4.6 × 150 mm column was used. Eluent system was A: methanol:water:tert-butyl methyl ether (TBME) 70:30:30 *v/v*%, while eluent B was methanol:TBME 50:50 *v/v*%. Solvents used were all at least HPLC grade and were purchased from VWR International (Radnor, Pennsylvania, United States). Absorbance was recorded at 250–700 nm with 1.2 nm resolution and 20 Hz with a PDA detector. Authentic standards were purchased (isomerized also to identify respective geometric isomers) from Merck-Sigma group. Matching retention order and absorption maximums have been reported under similar chromatographic conditions [79]. Quantitation was carried out at maximum absorbance against the internal standard. Further method details are available as Appendix A.

### 4.6. Sample Preparation and GC Metabolomics Analyses of β-Alanine and γ-Aminobutyric Acid

The 4D GCxGC TOFMS sample preparation is based on Canellas et al. [80] with modifications. Leaf and root samples (200 mg) were extracted after the internal standard (ribitol 30 µL of 1 mg ml^−1^ solution) was added, twice with 1 mL 60 *v/v* % methanol and twice with 1 mL 90 *v/v* % methanol. The extraction was done with vortex for 30 s, then 5 min in ultrasonic bath at room temperature. It was repeated and vortexed for 15 s followed by ultrasonic bath again at room temperature. Afterwards, the samples were centrifuged at 10,000× *g* for 5 min at 4 °C; the supernatant was collected and dried under vacuum. For derivatization, methoxyamine hydrochloride-dissolved pyridine (20 mg ml^−1^) (Merck-Sigma group, Darmstadt, Germany) was added to 200 µL of the extract and incubated at 37 °C for 90 min, which was followed by the addition of N-Trimethylsilyl-N-methyl trifluoroacetamide (Merck-Sigma group, Darmstadt, Germany) and incubation for 30 min at the same temperature.

The samples were transferred to vials and injected in split mode to the LECO Pegasus 4D GCxGC TOFMS (LECO Corp., St. Joseph, MI, USA) equipped with 30 m column (Rxi-5MS phase) and 1.5 m column (Rxi-17Sil MS phase). 1µL of the sample was injected to the column at 230 °C, the transfer line and ion source was at 250 °C. The carrier gas was He (Gruppo SIAD, Bergamo, Italy) and was used at constant flow rate (1 mL min^−1^). The thermal program started with 70 °C for 3 min, then it increased to 320 °C in 7 °C/minutes rate and maintained this high temperature for 5 min with 3.25 s modulation period in the 2D GC mode. For the identification standards and Kovats retention index were used. For data evaluation LECO ChromaTOF program (LECO Corp., St. Joseph, MI, USA) was used. Both the GC analyses and data processing were carried out by ChromaTOF 4.72 with Finn and Nist databases.

### 4.7. Gene Expression Analysis

For gene expression studies the second, fully developed leaves and roots of 14-day-old wheat plants were taken and immediately stored in liquid nitrogen. Total RNA extraction and cDNA synthesis were carried out according to Tajti et al. [81]. RT-qPCR measurements were performed on a BioRad CFX96 Touch Real-Time Detection System (Bio-Rad Laboratories, Inc., Hercules, CA, USA) using 1 µL 4-fold diluted cDNA, 200 nM forward and reverse primers, 2.5 µL PCRBIO Mastermix (PCR Biosystem Ltd., London, UK) and 2.5 µL molecular grade water. Relative transcript levels were determined with the 2^−ΔΔCt^ method [82]. For normalization Ta2291 was used as internal control gene. Primer sequences are available in Appendix A [83,84,85,86].

### 4.8. Statistical Analysis

Data are presented for the most representative repetition of the three independent biological experiments. The results are the means of at least three replicates for chromatographic determinations. The data were statistically evaluated using the standard deviation in Microsoft Excel. Different letters indicate statistically significant differences (*p* < 0.05) among multiple groups (one-way ANOVA with Duncan post hoc test was performed using SPSS 16.0).

## 5. Conclusions

Although the protective and roborative effects of exogenous PAs are demonstrated under several conditions in various plant species including their photoprotective role, the light-related influence on the metabolism of PAs is less understood. According to these, the main hypothesis of this experiment was that different light periods or light intensities (i.e., different light quantities) influence the PA pool and metabolism in different ways and modify the effect of exogenous PAs.

Based on the two experiments, light induced the accumulation of polyamines in the leaves mainly via the induction of PUT, in parallel with a decrease in DAP content (Figure 13).

Figure 14 summarizes the significant changes in the PA metabolism induced by light intensity with PA treatments in both the leaves and roots. The results show that although light intensity alone has remarkable effect on the PA pool, under the relevant light conditions (elevated, medium, or low light), a steady state condition is established, and thus the gene expression levels are not influenced. However, when exogenous PA treatments were applied, the excess of PA induced different steps of the PA metabolism depending on the light conditions. Leaves and roots responded differently, obviously since light has greater effect in the leaves, while PA treatments in the roots. In addition, the effect of light in the leaves was dominant at elevated, while the most pronounced influence of PAs, especially that of SPM, was found under low light conditions.

In the leaves, under elevated light conditions, due to the initially higher PUT and SPD content, PA treatments inhibited the synthesis of higher PAs but induced their back-conversion. Under medium light conditions, the gene expression of most enzymes involved in PA biosynthesis was inhibited. Furthermore, the *perPAO* expression also decreased, as no back-conversion is needed besides the increased PUT content. Under low light, PA treatments increased only the PUT content, leading to the inhibition of the PA synthesis and back conversion but induction of terminal catabolism. 

In the roots, under higher light intensity, the PA treatments induced the in vivo PA synthesis, and instead of the back conversion, PAs canalized towards the terminal catabolism. Although the gene expression of the PA metabolite enzymes did not change under medium light conditions, the PA pool and DAP content were still increased by the PA treatments. Nevertheless, under low light, PA treatments exert the highest influence on PA metabolism, as especially after SPM treatment—due to the increased PA and DAP levels—further activation of the terminal catabolism was found.

The sufficiently modulated PA metabolism together with the induction of SA and ABA signaling could be responsible for the observed positive effect of exogenous PAs under lower light conditions; however, depending on the light intensity, different PA metabolism-modulating strategies can be successful. It should also be taken into consideration that other PA metabolism-related processes can also be involved. For example, the role of polyamine transporters in plants has only been recognized recently. Besides the different effect of the individual PAs on PA uptake transporters, light conditions can also influence their expression or activity, which in turn modulate the accumulation and metabolism of PAs in PA-treated plants.

## Figures and Tables

**Figure 1 ijms-22-11717-f001:**
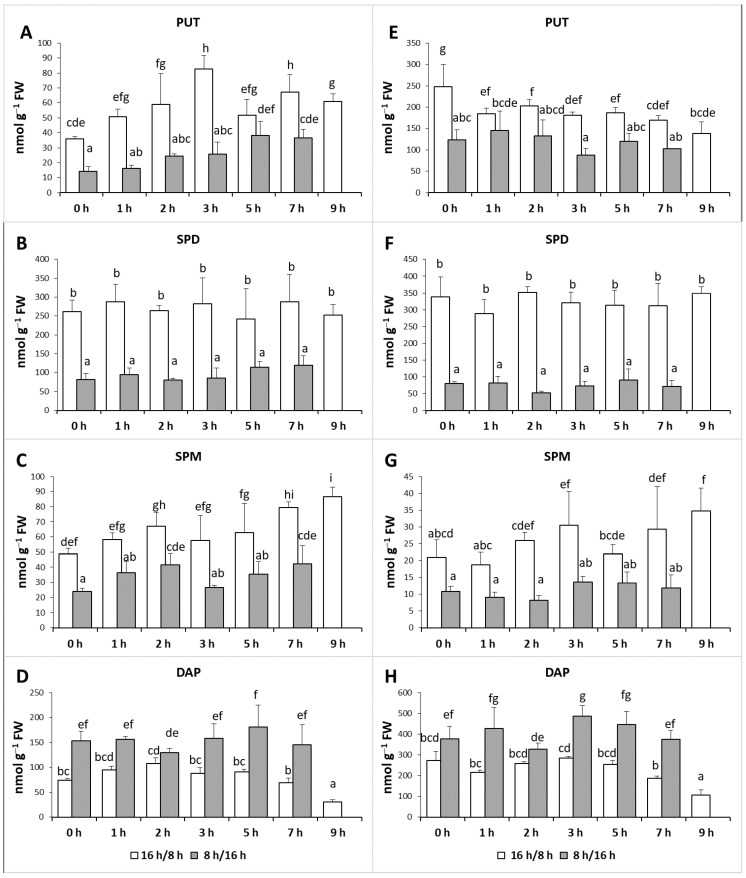
Effect of lighting hours on the polyamine patterns in the leaves (**A**–**D**) and roots (**E**–**H**) of plants grown under 16 h/8 h (white column) or 8 h/16 h day/night light period (dark grey column). PUT: putrescine, SPD: spermidine, SPM: spermine, DAP: 1,3-diaminopropane. Data represent mean values ± SD. Different letters indicate significant differences at *p* ≤ 0.05 level, among all the values of the given compounds.

**Figure 2 ijms-22-11717-f002:**
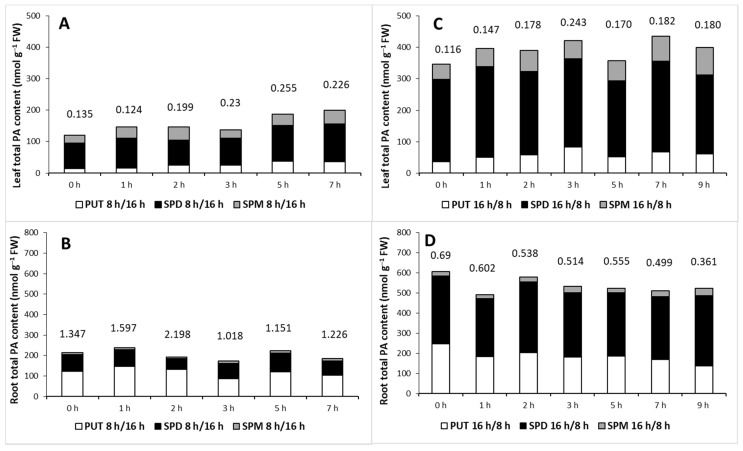
Effect of hours of illumination on the polyamine (PA) content in the leaves (**A**,**C**) and roots (**B**,**D**) of plants grown under 8 h/16 h (**A**,**B**) or 16 h/8 h (**C**,D) day/night light period. PUT: putrescine, SPD: spermidine, SPM: spermine. Data represent mean values ± SD. Numbers indicate the ratio of PUT/(SPD + SPM).

**Figure 3 ijms-22-11717-f003:**
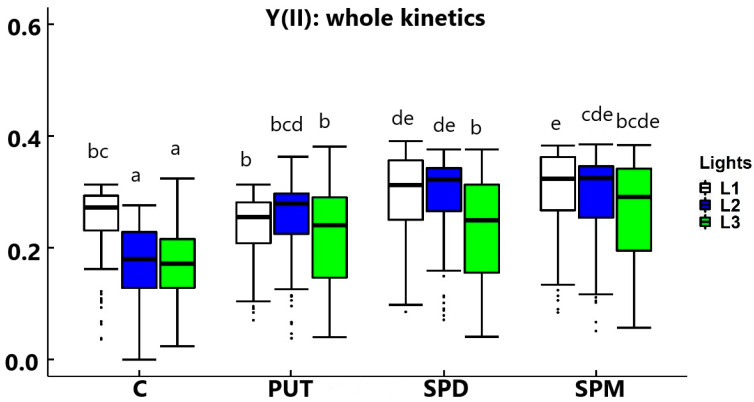
Box-plot presentation of the effect of different light intensities on the actual quantum yield of PSII [Y(II)] of plants grown under 16 h/8 h day/night light period with different light intensity conditions: 500 (elevated light: L1), 250 (medium light: L2) and 50 (low light: L3) µmol m^−2^ s^−1^ LED light after 7 days of 0.3 mM exogenous putrescine (PUT), spermidine (SPD) or spermine (SPM) treatments or without any treatment (C). Boxes represent Q1 and Q3 quartiles, the middle line is the median (Q2), and the whiskers show the minimum and maximum values. Different letters indicate significant differences at *p* ≤ 0.05 level, among all treatments.

**Figure 4 ijms-22-11717-f004:**
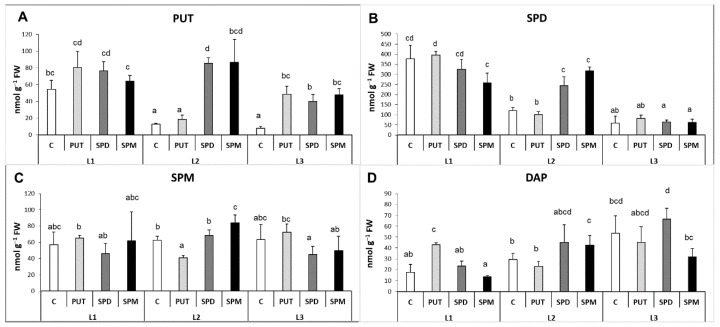
Effect of different light intensities on the polyamine pattern (**A**: PUT: putrescine, **B**: SPD: spermidine, **C**: SPM: spermine, **D**: DAP: 1,3-diaminopropane) in the leaves of plants grown under 16 h/8 h day/night light period with different light intensity conditions: 500 (elevated light: L1), 250 (medium light: L2) and 50 (low light: L3) µmol m^2^ s^−1^ LED light after 7 days of 0.3 mM exogenous putrescine (PUT), spermidine (SPD) or spermine (SPM) treatments or without any treatment (C). Data represent mean values ± SD. Different letters indicate significant differences at *p* ≤ 0.05 level, among all the values of the given compounds.

**Figure 5 ijms-22-11717-f005:**
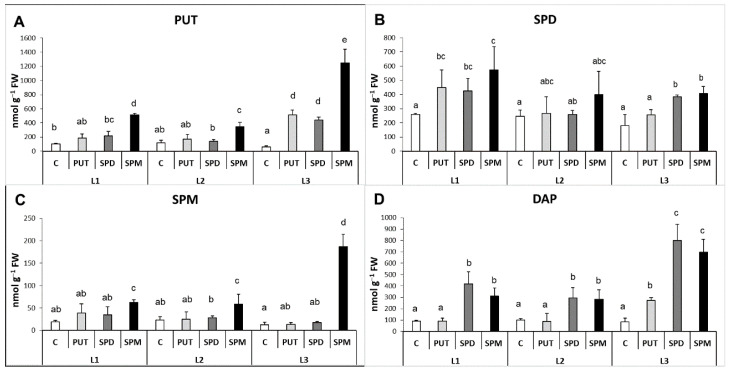
Effect of different light intensities on the polyamine pattern (**A**: PUT: putrescine, **B**: SPD: spermidine, **C**: SPM: spermine, **D**: DAP: 1,3-diaminopropane) in the roots of plants grown under 16 h/8 h day/night light period with different light intensity conditions: 500 (elevated light: L1), 250 (medium light: L2) and 50 (low light: L3) µmol m^2^ s^−1^ LED light after 7 days of 0.3 mM exogenous putrescine (PUT), spermidine (SPD) or spermine (SPM) treatments or without any treatment (C). PUT: putrescine, SPD: spermidine, SPM: spermine, DAP: 1,3-diaminopropane. Data represent mean values ± SD. Different letters indicate significant differences at *p* ≤ 0.05 level, among all the values of the given compounds.

**Figure 6 ijms-22-11717-f006:**
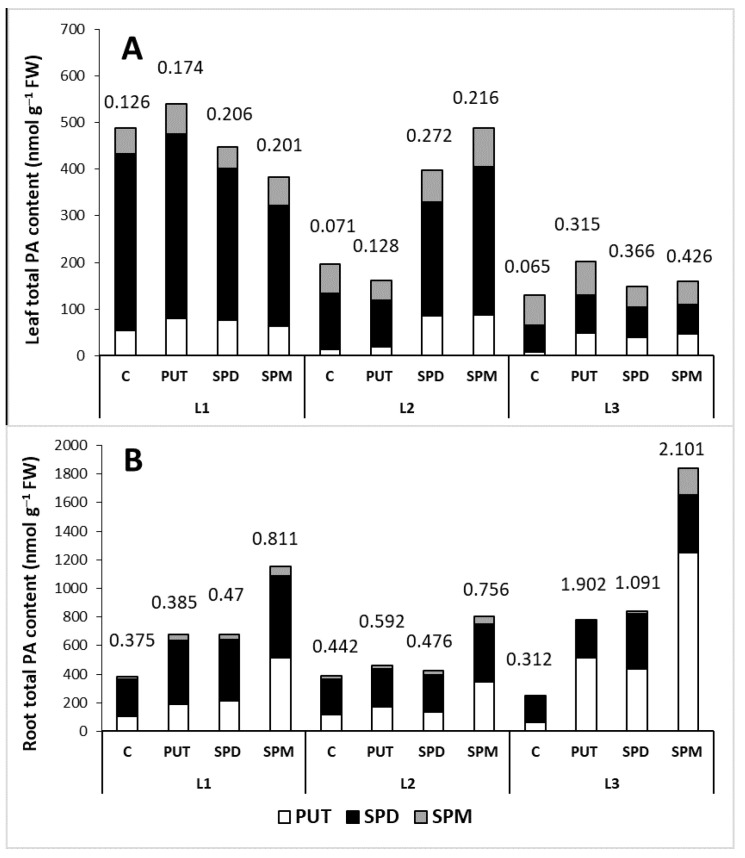
Effect of light intensity on the total polyamine (PA) content in the leaves (**A**) and roots (**B**) of plants grown under 16 h/8 h day/night light period with different light intensity conditions: 500 (elevated light: L1), 250 (medium light: L2) and 50 (low light: L3) µmol m^2^ s^−1^ LED light after 7 days of 0.3 mM exogenous putrescine (PUT), spermidine (SPD) or spermine (SPM) treatments or without any treatment (C). Numbers indicate the ratio of PUT/(SPD + SPM).

**Figure 7 ijms-22-11717-f007:**
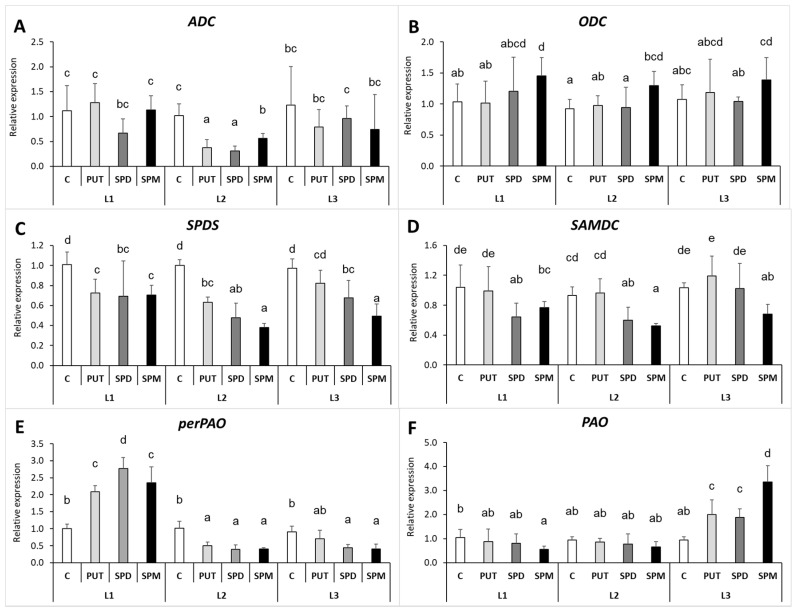
Effect of different light intensities on the gene expression patterns of arginine decarboxylase *(***A***: ADC*), ornithine decarboxylase (**B**) (*ODC*), spermidine synthase (**C**) (*SPDS*), S-adenosylmethionine decarboxylase (**D**) (*SAMDC*), peroxisomal polyamine oxidase (**E**) (*perPAO*) and polyamine oxidase (**F**) (*PAO*) in the leaves of plants grown under 16 h/8 h day/night light period with different light intensity conditions: 500 (elevated light: L1), 250 (medium light: L2) and 50 (low light: L3) µmol m^2^ s^−1^ LED light after 7 days of 0.3 mM exogenous putrescine (PUT), spermidine (SPD) or spermine (SPM) treatments or without any treatment (C). Data represent mean values ± SD. Different letters indicate significant differences at *p* ≤ 0.05 level, among all the values of the given compounds.

**Figure 8 ijms-22-11717-f008:**
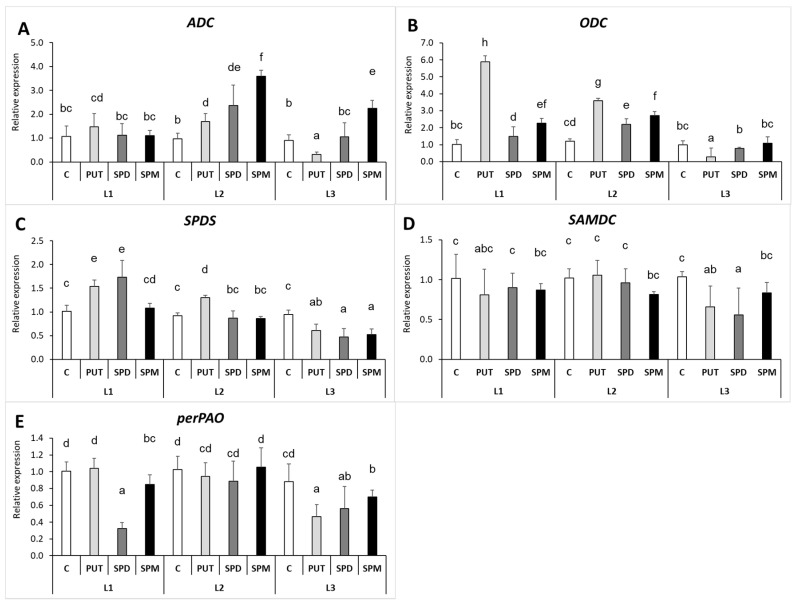
Effect of different light intensities on the gene expression patterns of arginine decarboxylase *(***A**) (*ADC*), ornithine decarboxylase (**B**) (*ODC*), spermidine synthase (**C**) (*SPDS*), S-adenosylmethionine decarboxylase (**D**) (*SAMDC*), peroxisomal and polyamine oxidase (**E**) (*perPAO)* in the roots of plants grown under 16 h/8 h day/night light period with different light intensity conditions: 500 (elevated light: L1), 250 (medium light: L2) and 50 (low light: L3) µmol m^2^ s^−1^ LED light after 7 days of 0.3 mM exogenous putrescine (PUT), spermidine (SPD) or spermine (SPM) treatments or without any treatment (C). Data represent mean values ± SD. Different letters indicate significant differences at *p* ≤ 0.05 level, among all the values of the given compounds.

**Figure 9 ijms-22-11717-f009:**
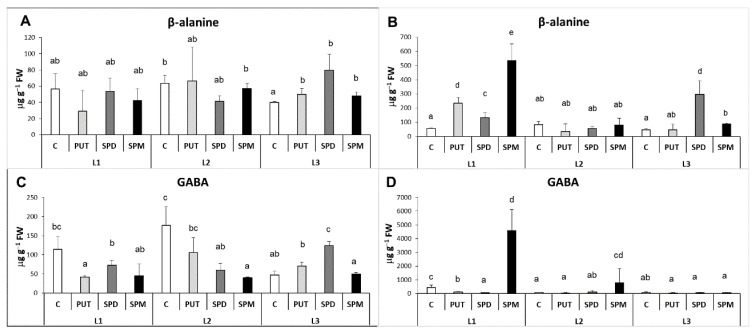
Effect of different light intensities on the levels of β-alanine (**A**,**B**) and γ-aminobutyric acid (GABA) (**C**,**D**) in the leaves (**A**,**C**) and roots (**B**,**D**) of plants grown under 16 h/8 h day/night light period with different light intensity conditions: 500 (elevated light: L1), 250 (medium light: L2) and 50 (low light: L3) µmol m^2^ s^−1^ LED light after 7 days of 0.3 mM exogenous putrescine (PUT), spermidine (SPD) or spermine (SPM) treatments or without any treatment (C). PUT: putrescine, SPD: spermidine, SPM: spermine, GABA: γ-aminobutyric acid. Data represent mean values ± SD. Different letters indicate significant differences at *p* ≤ 0.05 level, among all the values of the given compounds.

**Figure 10 ijms-22-11717-f010:**
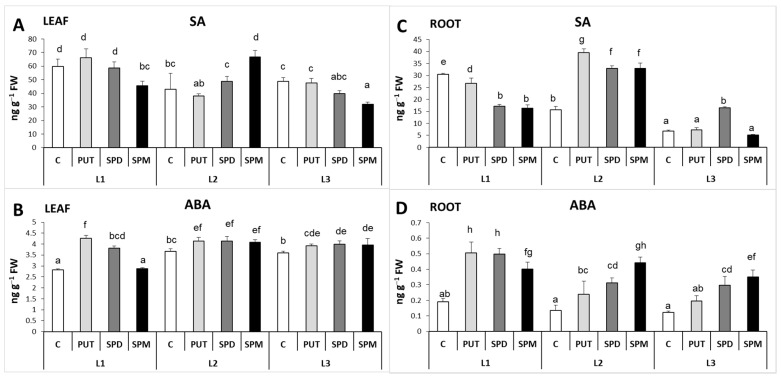
Effect of different light intensities on the salicylic acid (SA) (**A**,**C**) and abscisic acid (ABA) (**B**,**D**) content in the leaves (**A**,**B**) and roots (**C**,**D**) of plants grown under 16 h/8 h day/night light period with different light intensity conditions: 500 (elevated light: L1), 250 (medium light: L2) and 50 (lower light: L3) µmol m^2^ s^−1^ LED light after 7 days of 0.3 mM exogenous putrescine (PUT), spermidine (SPD) or spermine (SPM) treatments or without any treatment (C). Data represent mean values ± SD. Different letters indicate significant differences at *p* ≤ 0.05 level, among all the values of the given compounds.

**Figure 11 ijms-22-11717-f011:**
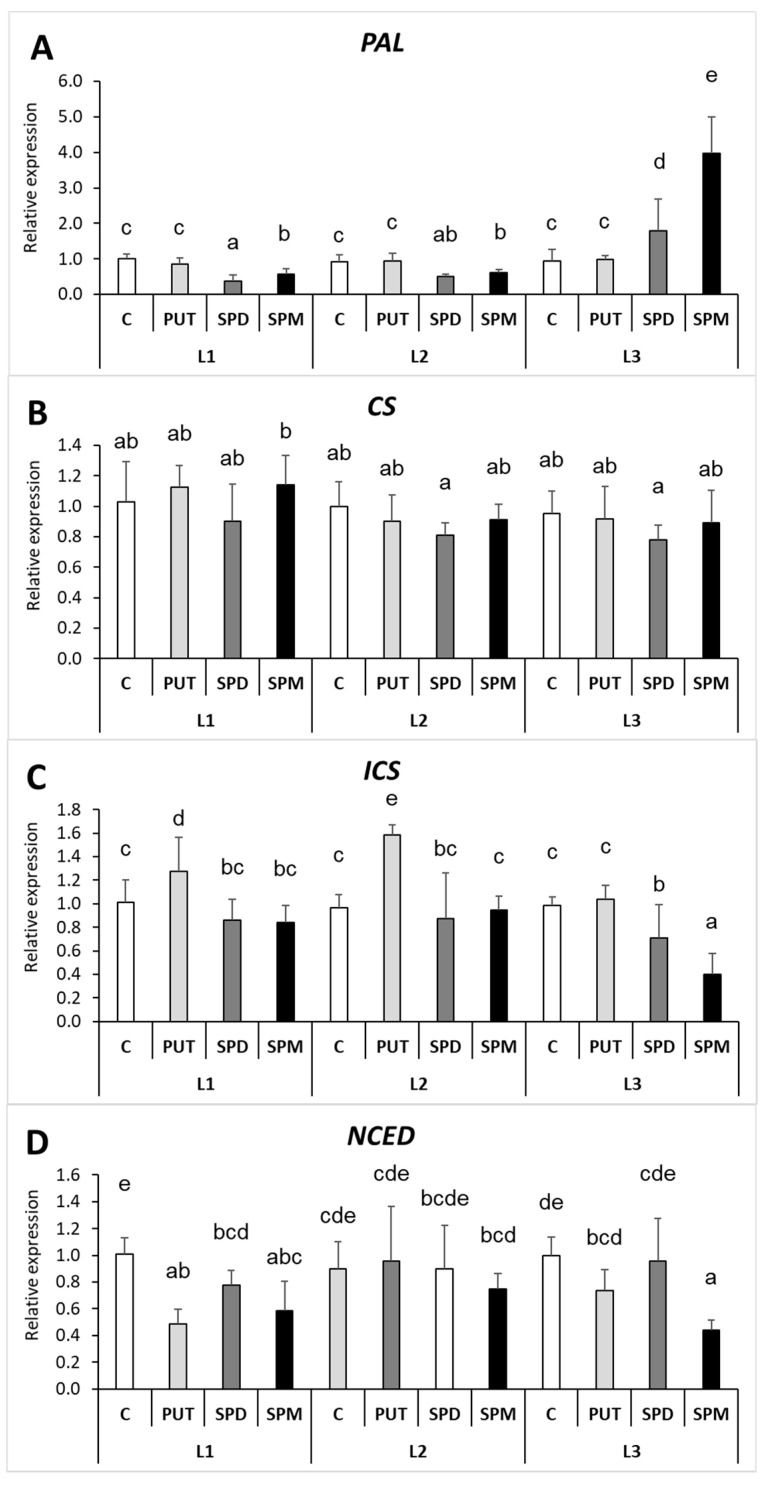
Effect of different light intensities on the gene expression levels of phenylalanine-ammonia-lyase (**A**) (*PAL*), chorismate synthase (**B**) (*CS*), isochorismate synthase (**C**) (*ICS*) and 9-cis-epoxycarotenoid dioxygenase (**D**) (*NCED*) in the leaves of plants grown under 16 h/8 h day/night light period with different light intensity conditions: 500 (elevated light: L1), 250 (medium light: L2) and 50 (low light: L3) µmol m^2^ s^−1^ LED light after 7 days of 0.3 mM exogenous putrescine (PUT), spermidine (SPD) or spermine (SPM) treatments or without any treatment (**C**). Data represent mean values ± SD. Different letters indicate significant differences at *p* ≤ 0.05 level, among all the values of the given compounds.

**Figure 12 ijms-22-11717-f012:**
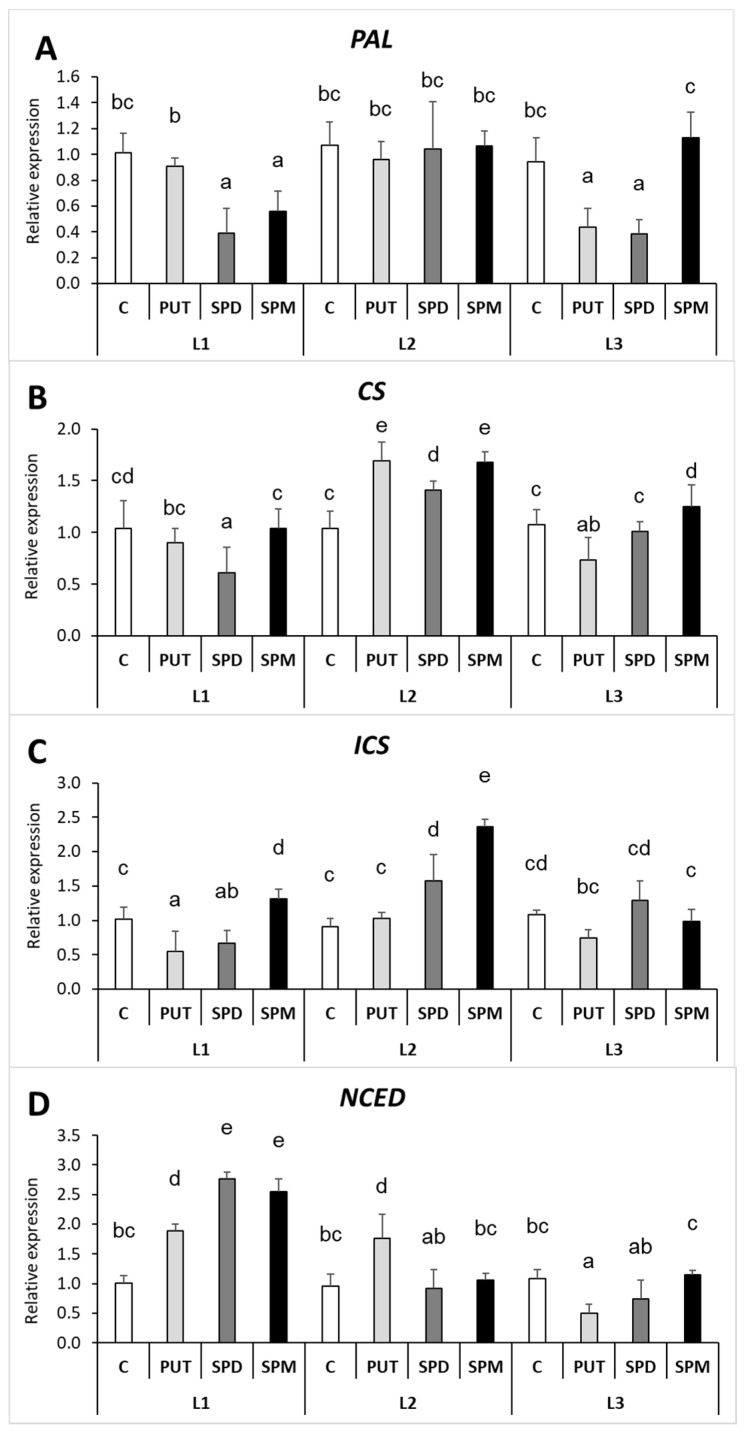
Effect of different light intensities on the gene expression levels of phenylalanine-ammonia-lyase (**A**) (*PAL*), chorismate synthase (**B**) (*CS*), isochorismate synthase (**C**) (*ICS*) and 9-cis-epoxycarotenoid dioxygenase (**D**) (*NCED*) in the roots of plants grown under 16 h/8 h day/night light period with different light intensity conditions: 500 (elevated light: L1), 250 (medium light: L2) and 50 (low light: L3) µmol m^2^ s^−1^ LED light after 7 days of 0.3 mM exogenous putrescine (PUT), spermidine (SPD) or spermine (SPM) treatments or without any treatment (C). Data represent mean values ± SD. Different letters indicate significant differences at *p* ≤ 0.05 level, among all the values of the given compounds.

**Figure 13 ijms-22-11717-f013:**
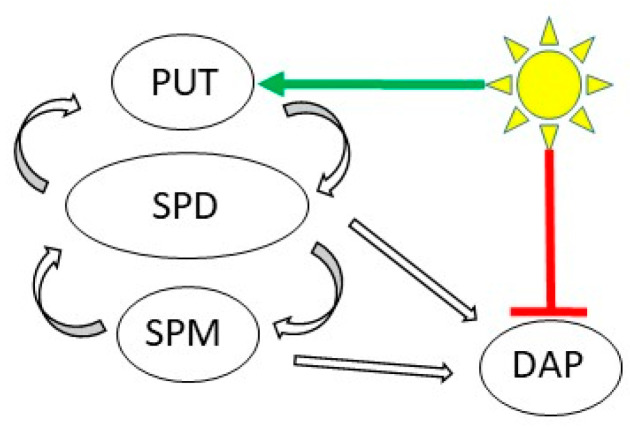
Schematic representation of the effects of light on polyamine pool. Light induces the putrescine (PUT) level first and it also reduces the 1,3-diaminopropane (DAP) content. (SPD: spermidine, SPM: spermine).

**Figure 14 ijms-22-11717-f014:**
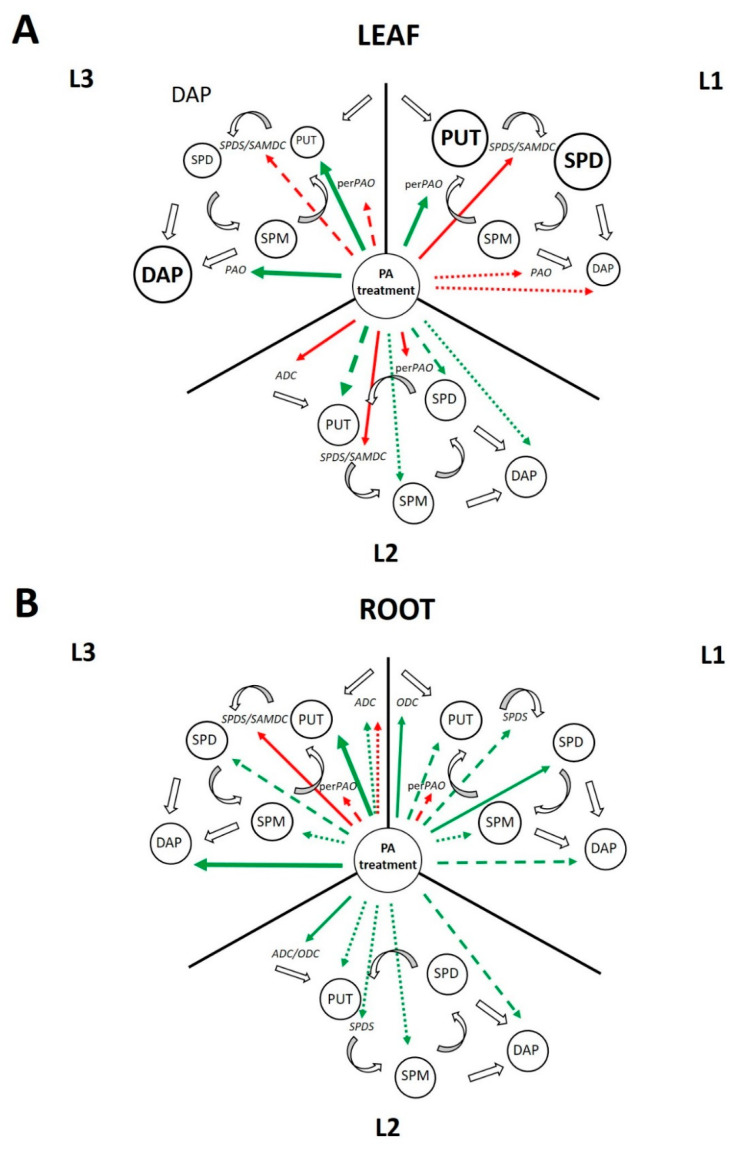
Significant changes in the polyamine metabolism in the leaves (**A**) and roots (**B**) of wheat plants induced by light intensity (L1: elevated, L2: medium and L3: low light conditions). The size of the circles represents light-induced differences in polyamine (PA) contents (PUT: putrescine, SPD: spermidine, SPM: spermine, DAP: 1,3-diaminopropane). The gene expression levels were not influenced by the light intensity alone. Green arrows indicate increasing, while red arrows indicate decreasing effect of the polyamine treatments on polyamine contents or gene expression levels. Solid arrows indicate that all the polyamine treatments have the same effect; dashed arrows show that only 2 of the applied polyamine treatments have the same effect, while dotted arrows represent that only one of the polyamine treatments has influence. Thickness of the arrows indicates the strength of the effects. For details, see the text in the conclusion.

**Table 1 ijms-22-11717-t001:** Changes in the pigment contents during the illumination period of 7 h or 9 h in plants grown under 8 h/16 h or 16 h/8 h day/night light period. The sampling at 0 h means that plants were sampled in the dark, directly before the onset of the light. Data represent mean values ± SD. Different letters indicate significant differences at *p* ≤ 0.05 level, among all the values of the given compounds.

Content (μg g^−1^ FW)		Trans- Violaxanthin	Trans- Neoxhantin	Trans- Lutein	Chlorophyll a	Chlorophyll b	Pheophytin a	Trans- β-carotene	9-Cis- β-carotene
8h /16 h	0 h	45.77 ± 1.1 a	43.23 ± 0.67 a	146.33 ± 2.89 a	480.67 ± 9.29 a	241.67 ± 4.16 a	10.87 ± 0.15 e	48.53 ± 0.85 a	4.65 ± 0.11 a
	1 h	55.33 ± 1.36 bcd	53.27 ± 0.83 ef	167.33 ± 2.31 d	543.33 ± 5.77 b	270.33 ± 2.31 cde	11.8 ± 0.17 e	55.83 ± 1.24 efg	4.66 ± 0.07 a
	2 h	55.3 ± 0.9 efg	50.63 ± 0.75 def	164.67 ± 2.52 d	547 ± 5.29 b	268.67 ± 2.31 bcd	11.13 ± 0.06 c	52.17 ± 1.79 b	4.74 ± 0.13 ab
	3 h	59.17 ± 0.67 g	55.2 ± 0.26 g	172.33 ± 1.53 ef	561 ± 7.21 cd	275 ± 3.61 e	10.87 ± 0.15 e	53.17 ± 1.66 bcd	4.95 ± 0.09 b
	5 h	57.8 ± 0.98 bcd	52.17 ± 0.81 d	166 ± 2 cd	544.67 ± 5.69 bc	267 ± 2.65 bcde	9.56 ± 0.12 e	51.63 ± 1.25 bc	4.76 ± 0.05 ab
	7 h	58.9 ± 1.08 fg	53.2 ± 0.89 ef	169 ± 1.73 de	555.33 ± 4.04 bcd	273 ± 1.73 de	9.68 ± 0.13 c	53.53 ± 0.6 bcd	4.79 ± 0.09 ab
16 h/8 h	0 h	66.23 ± 2.08 h	55.57 ± 1.72 g	187 ± 4.58 g	631.69 ± 8.74 g	292.69 ± 4.16 f	9.96 ± 0.26 d	62.2 ± 1.92 h	6.04 ± 0.11 e
	1 h	56.17 ± 0.97 cde	46.27 ± 0.91 bc	166.33 ± 2.89 d	584 ± 11.27 f	273.67 ± 5.77 de	8.74 ± 0.13 ab	56.83 ± 1.1 fg	4.91 ± 0.12 b
	2 h	54.5 ± 0.62 bc	45.63 ± 0.86 b	160.33 ± 1.53 bc	562.67 ± 3.06 de	263.66 ± 1.15 bc	8.67 ± 0.09 ab	54.27 ± 1.38 cde	4.66 ± 0.3 a
	3 h	64.73 ± 0.4 h	53.93 ± 0.51 fg	182.67 ± 2.08 g	625.33 ± 7.51 g	292.67 ± 3.51 f	9.61 ± 0.12 c	63.23 ± 0.25 h	5.21 ± 0.13 c
	5 h	53.93 ± 0.51 b	44.73 ± 0.23 ab	158.67 ± 1.15 b	561.33 ± 4.93 cd	262.67 ± 3.06 b	8.65 ± 0.1 ab	55.1 ± 0.56 def	5.15 ± 0.08 c
	7 h	56.9 ± 0.87 def	47.96 ± 1.4 c	165.67 ± 2.52 d	578 ± 6f	272.67 ± 3.06 de	8.85 ± 0.08 b	57.37 ± 0.45 g	5.32 ± 0.03 c
	9 h	57.77 ± 1.72 efg	51.8 ± 1.85 de	175.67 ± 5.51 f	575.67 ± 15.57 ef	273 ± 7.55 de	8.48 ± 0.27 a	57.53 ± 1.42 g	5.63 ± 0.07 d

**Table 2 ijms-22-11717-t002:** Changes in the pigment contents under different light intensity conditions: 500 (elevated light: L1), 250 (medium light: L2) and 50 (low light: L3) µmol m^2^ s^−1^ LED light after 7 days of 0.3 mM exogenous putrescine (PUT), spermidine (SPD) or spermine (SPM) treatments or without any treatment. Data represent mean values ± SD. Different letters indicate significant differences at *p* ≤ 0.05 level, among all the values of the given compounds.

Content (μg g^−1^ FW)		Trans- Violaxhantin	Trans-Neoxanthin	Trans- Lutein	Chlorophyll a	Chlorophyll b	Pheophytin a	Trans-β-carotene	9-Cis-β-carotene
L1	C	76.83 ± 1.59 f	58.03 ± 2.65 c	208 ± 7 f	716.33 ± 25.66 e	307 ± 11.79	10.7 ± 0.4	72.67 ± 1.19 f	7.39 ± 0.1 e
	PUT	65.33 ± 4.97 cde	50.57 ± 4.33 bc	178.67 ± 11.68 de	594.33 ± 31.56 cd	253 ± 11.53	9.48 ± 0.47	69.9 ± 2.86 f	6.46 ± 0.17 cd
	SPD	73.57 ± 5.95 ef	51.8 ± 3.87 bc	189.33 ± 14.98 e	639 ± 55.34 d	273 ± 24.56	9.4 ± 0.92	66.6 ± 1.28 ef	6.87 ± 0.45 d
	SPM	70.4 ± 2.19 def	53.73 ± 1.91 bc	187.67 ± 4.73 e	624.67 ± 14.19 cd	264.33 ± 5.51 cd	9.67 ± 0.23 def	70 ± 1.4 f	6.7 ± 0.14 d
L2	C	57.67 ± 0.42 bc	52.43 ± 0.46 bc	189 ± 1 e	639.67 ± 3.05	280.67 ± 1.1	8.7 ± 0.04 cdef	59.27 ± 0.67 de	6.42 ± 0.11 cd
	PUT	56.93 ± 0.86 bc	52.17 ± 0.25 b	184.33 ± 3.51 e	606.67 ± 13.87	266 ± 6.24	8.49 ± 0.26 bcde	58.67 ± 1.67 de	6.06 ± 0.15 bc
	SPD	63.93 ± 5.86 cd	54.17 ± 5.95 bc	192.3 ± 16.01 ef	617 ± 51.1	274 ± 23.9	10.06 ± 3.51 ef	60.17 ± 4.64 de	6.32 ± 0.39 cd
	SPM	54.93 ± 12.08 b	50.27 ± 6.67 b	185 ± 12.49 e	610.33 ± 40.65	273.3 ± 11.72	8.49 ± 0.76 bcde	56.73 ± 6.92 cd	5.98 ± 0.57 bc
L3	C	39.83 ± 0.74 a	41.07 ± 0.38 a	159 ± 2 ab	508 ± 16.1	233 ± 7.21	6.529 ± 0.23 ab	45.67 ± 0.14 ab	5.05 ± 0.07 a
	PUT	44.3 ± 1.57 a	44.27 ± 1.58 a	174.33 ± 4.51 cde	566.67 ± 10.41	262.67 ± 4.16	7.71 ± 0.16 abcd	50.33 ± 0.86 bc	5.61 ± 0.16 b
	SPD	40.47 ± 3.7 a	42.57 ± 4.64 a	162.67 ± 14.64 abc	517.67 ± 39.01	239 ± 17	7.09 ± 0.6 abc	41.1 ± 12.99 ab	5.09 ± 0.53 a
	SPM	36.5 ± 0.26 a	39.63 ± 1.11 a	151 ± 3a	467.33 ± 9.02	218 ± 4.58	6.04 ± 0.13 ab	43.83 ± 0.57 ab	4.81 ± 0.04 a

**Table 3 ijms-22-11717-t003:** Correlation analysis on the polyamine, salicylic acid and abscisic acid contents in the leaves of wheat plants. Correlations significant at 0.05 are highlighted in bold. DAP: 1,3-diaminopropane; PUT: putrescine, SPD: spermidine, SPM: spermine, SA: salicylic acid; ABA: abscisic acid.

	PUT	SPD	SPM	DAP	SA	ABA
PUT	1	**0.752**	0.387	−0.075	0.618	0.14
SPD		1	0.282	−0.389	**0.85**	−0.202
SPM			1	0.205	0.538	0.04
DAP				1	−0.03	0.565
SA					1	−0.022
ABA						1

**Table 4 ijms-22-11717-t004:** Correlation analysis on the polyamine, salicylic acid and abscisic acid contents in the roots of wheat plants. Correlations significant at 0.05 are highlighted in bold. DAP: 1,3-diaminopropane; PUT: putrescine, SPD: spermidine, SPM: spermine, SA: salicylic acid; ABA: abscisic acid.

	PUT	SPD	SPM	DAP	SA	ABA
PUT	1	0.432	**0.882**	**0.704**	−0.501	0.227
SPD		1	0.443	0.415	−0.034	**0.823**
SPM			1	0.506	−0.303	0.344
DAP				1	−0.388	0.302
SA					1	0.216
ABA						1

## Data Availability

Data are involved in the study.

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
