# Peer review of "Polyamine Metabolism under Different Light Regimes in Wheat"

_ijms, 2021, doi:10.3390/ijms222111717_

Round 1

Reviewer 1 Report

The manuscript presents new information about polyamines metabolism in wheat plants grown under different light regimes (intensity or duration) with or without presence of polyamines in nutrition media. The topic is interesting and could provoke attention in the researchers. However the manuscript suffers from some faults that should be corrected.

Introduction part should be more focused to the aim of the research.

It is not enough clear why the authors decided to measure polyamine levels (and respective genes involved into polyamine metabolic pathway), other phytohormones, and metabolites exactly after 4 h of illumination (second model system). They stated that their choice is based on the results of the first experiment, but the parameters in this model system were measured at 3 or 5 h of illumination? How different levels of polyamines detected in leaves and roots (Figure 1) could be explained (see y-axis ranges)? Table 1 and Table 2 should be placed landscape, because it is not possible to see all data. How the different content of polyamines measured 3-5 h of illumination during the first experiment (in plants grown under 190 μmol m-2 s-1 - Figure 1) and those measured during the second experiment (in plants grown under L2 regime - 250 μmol m-2 s-1 – Figures 3 and 4) could be explained in controls? The illumination is almost the same but control levels of Put, Spd, and DAP differed 2-6 times. Only control Spm level is nearly equal in both experimental models. How the increase of Spd and Put due to PAs application in L2-grown plants could be explained together with the decrease in expression levels of ADC, SPDS, and SAMDC genes, and without significant change of ODC? Results of NCED expression in roots were not described at all. Tables (or Figures) of total PAs content as well ratio Put/Spd+Spm are not given although such results are described on p. 7 of the manuscript and discussed later. They should be included. Discussion part should be rewritten and to be more focused on the topic of the research. In the present form it seems like second introduction with repetition of the results. SA-PAs connections, and especially these sentences about exogenous SA are not necessary (exclude them with the relevant references 65-68).

Minor comments:

The word ‘question’ in the abstract is not suitable – change with aim, goal or purpose of the study.

Change under “control” conditions with under “optimal growth” conditions (second paragraph of Introduction). Also “adequate” controls change with “respective”, “corresponding” or “relevant” throughout the manuscript.

Please check very carefully English grammar.

In Figure 6 legend change “leaves” with “roots”, in Figure 7 legend panels of roots are B and D – correct it.

Omit “Kami” from 4.5 sub-division title.

Omit “Kinga” from 4.6 sub-division title, and define the metabolites which you measured. Did you really use only 0.2 mg plant material for determination of polyamines? Is centrifugation 1,4 000 g or 14 000 g for pigments?

Author Response

We would like to express our thanks to the reviewers for drawing our attention to any inaccuracies. All the criticism was constructive, which helped to optimize our study. We are resubmitting our MS after major revision, based on the reviewers' comment. Corrections in the main text are indicated in proofreading. Changes in the MS in response to each comment in the answers are also indicated. 

We hope the reviewer will find the new version more acceptable from publication.

Please see the attached file, as in this box figures and table maybe not appeared properly.

Reviewer 1:

The manuscript presents new information about polyamines metabolism in wheat plants grown under different light regimes (intensity or duration) with or without presence of polyamines in nutrition media. The topic is interesting and could provoke attention in the researchers. However the manuscript suffers from some faults that should be corrected.

Introduction part should be more focused to the aim of the research.

>>> The Introduction has been shortened and re-arranged in order be more focused on the aim of the present study.

It is not enough clear why the authors decided to measure polyamine levels (and respective genes involved into polyamine metabolic pathway), other phytohormones, and metabolites exactly after 4 h of illumination (second model system). They stated that their choice is based on the results of the first experiment, but the parameters in this model system were measured at 3 or 5 h of illumination?

>>> In case of the first experiment the main aim was to monitor changing in the polyamine contents in plants adapted to two different light/dark periods, and reveal the daily fluctuation. In this experiment, we sampled for gene expression, metabolite analyses immediately at the relevant hours, and it takes approximately 10 minutes. However, in the second experiment, where plants were grown under three different light regimes, with or without polyamine treatments, the sampling needs more time, approximately 1 hour. According to these, the sampling for the second experiment takes approximately 1 hour, from the 4th hour of illumination until the 5th hour of the illumination. The order of the sampling was: controls of L1-L3, PUT treatment from L1-L3, SPD treatment from L1-L3, and SPM treatment from L1-L3. To make it clear, we modified the Materials and methods:

“The sampling of plants was started after 4 h of illumination, as this time interval is sufficient for PA contents to reach a medium level during the day.”

How different levels of polyamines detected in leaves and roots (Figure 1) could be explained (see y-axis ranges)?

>>> Based on the literature, the levels of PAs changes dynamically, depending on the plant development phase, the plant species and also on the genotype. However, our previous results (Pál et al., 2014 J Hazardous Materials; Kovács et al. 2014 Environmental and Experimental Botany; Szalai et al., 2018 Plant Physiol Biochem; Darko et al., 2019 PlosOne; Pál et al., 2019 Plant Physiol Biochem) showed that the level of PUT is usually higher in the roots than in the leaves, the amount of SPD and SPM can be higher or equal in the leaves than that of the roots.

Table 1 and Table 2 should be placed landscape, because it is not possible to see all data.

>>> Table 1 and 2 were placed in landscape orientation in the first submittion, but the IJMS template re-arranged it. In order to see all the data, we exchanged rows and columns. We hope it is acceptable now.

How the different content of polyamines measured 3-5 h of illumination during the first experiment (in plants grown under 190 μmol m-2 s-1 - Figure 1) and those measured during the second experiment (in plants grown under L2 regime - 250 μmol m-2 s-1 – Figures 3 and 4) could be explained in controls? The illumination is almost the same but control levels of Put, Spd, and DAP differed 2-6 times. Only control Spm level is nearly equal in both experimental models.

>>> Although the absolute polyamine values were in the same range, the reviewer is right, there were some differences in the control values. These experiments were separated in time, which may also often cause some differences in the absolute values. Furthermore, one of the possible other reasons for this is, that although in both experiments plants were grown in the same type of growth chambers (Conviron PGR-15 plant growth chamber, Controlled Environments Ltd, Winnipeg, Canada), in the first experiment, where the aim was to monitor the daily changes in PA contents was performed, the growth chambers  used was equipped with High intensity discharge (HID) metal-halide lamps, similarly as in most of our previous studies (Pál et al. 2005; Physiol. Plantarum; Pál et al., 2014 J Hazardous Materials; Kovács et al. 2014 Environmental and Experimental Botany; Szalai et al., 2018 Plant Physiol Biochem; Pál et al., 2019 Plant Physiol Biochem). This type of metal halide lamps have relatively high fluence (max. 200 lumens per watt) and high photosynthetically active radiations (PARs) efficiency (max. 40%), and they are typically used in greenhouses and plant growth rooms.

However, the LED technology is predicted to replace fluorescent and HID lamps in horticultural systems and to revolutionize controlled growth environments.  So in the second experiment, a chamber of the same type, but equipped with a continuous wide spectrum LED (Philips Lumileds) LXZ2-5790-y) was used, as it was indicated in the material and methods, according to Gyugos et al., 2019 J Agron Crop Sci).

Here in the table the comparison of the fractions and ratios of the different light sources reveals that in the chamber with HID lamps, the ratio of far-red is higher than the chamber with LED light sources:

Treatments

Blue %

Green %

Red %

Far-red %

Blue/Red

Red/Far-red

blue/∑

Red/∑

Far-red/∑

Blue/Far-red

L1

21.435

33.805

43.248

1.51

0.50

21

0.201923

0.403846

0.019231

10.5

L2

21.013

34.521

42.026

2.439

0.50

17.23077

0.210131

0.420263

0.02439

8.615384615

L3

20.19

37.5

40.384

1.923

0.50

28.625

0.214353

0.432483

0.015109

14.1875

HID

16.136

39.809

32.909

11.146

0.49

2.9523

0.161359

0.329087

0.111465

1.447619

According to these differences in the light source of the chambers, the polyamine contents can show some differences, the absolute values cannot be compared directly, but the levels are still in the range of values normally experienced. Based on the different conditions we discussed the two experiment separately, but we concluded that in both experiments the light quantity (more illumination hours or higher intensity) influenced the PA content similarly, especially in case of PUT and DAP contents.

How the increase of Spd and Put due to PAs application in L2-grown plants could be explained together with the decrease in expression levels of ADC, SPDS, and SAMDC genes, and without significant change of ODC?

>>> Sampling for the determination of PA contents and the gene expression measurements were performed on the same day and hour. As the highest accumulation of PUT and SPD in proportion was found in the leaves under L2 conditions compared to the relevant control (adapted for a lower PA content due to the light regime), after a certain accumulation, the increased endogenous PA content as a feed-back mechanism can inhibit the in vivo PA synthesis, so the down-regulation is understandable. The difference between ADC and ODC expressions, could be explained by the fact, that the ADC pathway for PUT synthesis is mainly induced under stress conditions, while the ODC pathway is responsible for plant growth and development, according to these, this latter one has more stable expression.

Results of NCED expression in roots were not described at all.

>>> Thank you for the notification the 2.2.3 result section has been completed:

“The PA treatment-induced changes in the expression level of NCED compared to the relevant control in the roots showed correlation with the root ABA content only under L1 conditions, as both the ABA and the NCED transcript levels increased after all the PA treatments (Figure 10D).”

Tables (or Figures) of total PAs content as well ratio Put/Spd+Spm are not given although such results are described on p. 7 of the manuscript and discussed later. They should be included.

>>> The review is right. According to the suggestion of the reviewer, these parts on page 7 and page 11 about total PA and the ratio of PUT/(SPD+SPM) have been completed in the text and two additional figures:

On page 7:

“These results revealed that although the ratio of PAs in the leaves was different under the two hours of light/day conditions, but the changes in the PUT/(SPD+SPM) ratio mostly showed similar patterns during the day starting from the onset of illumination, as in-creased under both conditions. However, in the roots this ratio decreased only under longer illumination per day (Figure 2.)”

And on page 10:

“Overall, positive correlation was found between the light intensity and total PA contents of the leaves. In addition, it seems like the highest accumulation inducing effect of the PA treatments could be detected under L2 in the leaves, while under L3 conditions in the roots, compared to the relevant control. Nevertheless, the highest PUT/(SPD+SPM) ratio was found after PA treatments under L3 conditions both in the leaves and roots (Figure 6.).”

Discussion part should be rewritten and to be more focused on the topic of the research. In the present form it seems like second introduction with repetition of the results. SA-PAs connections, and especially these sentences about exogenous SA are not necessary (exclude them with the relevant references 65-68).

>>> The discussion section has been shortened, in several cases re-written in order to more focus on the main and important findings on PA metabolism, and to highlight the changes and differences presented on Figure 14. The unnecessary pieces of information about SA and SA-PA connection have been omitted. We hope that the revised discussion passes the massage of the MS.

Minor comments:

The word ‘question’ in the abstract is not suitable – change with aim, goal or purpose of the study.

Change under “control” conditions with under “optimal growth” conditions (second paragraph of Introduction). Also “adequate” controls change with “respective”, “corresponding” or “relevant” throughout the manuscript.

>>> Thank you for these suggestions, terms have been corrected according to your comment in text.

Please check very carefully English grammar.

>>> The manuscript has been reviewed linguistically by an English proofreader, changes are also indicated in the revision.

In Figure 6 legend change “leaves” with “roots”, in Figure 7 legend panels of roots are B and D – correct it.

>>> The legends of Figure 6 and 7 have been corrected.

Omit “Kami” from 4.5 sub-division title.

Omit “Kinga” from 4.6 sub-division title, and define the metabolites which you measured.

>>> The titles of section 4.5 and 4.6 have been corrected, thank you for the notification.

Did you really use only 0.2 mg plant material for determination of polyamines? Is centrifugation 1,4 000 g or 14 000 g for pigments?

>>> Sorry for the inaccuracies and typing errors. The title of subsection 4.6 corrected according to indicate which metabolites have been measured: 

“Sample preparation and GC metabolomics analyses of β-alanine and γ-aminobutyric acid”.

0.2 mg has been corrected, as 200 mg was used for polyamine determination.

1,4 000 g has been also corrected to 14,000.

Reviewer 2 Report

The manuscript that was submitted for analysis contains some interesting results. The experiments were applied correctly. I would like to express my appreciation for the Authors for a comprehensive approach to research. Various research methods and techniques have been applied, through biochemical methods and to plant physiology. It is very important and definitely facilitates the interpretation. The whole argument is conducted in a coherent and logical way, which makes it easier for the reader to follow the presented results and argue their interpretation. 

I have no criticisms of this manuscript. I only ask the Authors to remove their initials from the methodology :) I mean 4.5. ‘Kami’ pigment, please change to Pigments. ‘King’ should be removed 4.6. Sample preparation and GC metabolomics analyzes Kinga

Author Response

We would like to express our thanks to the reviewers for drawing our attention to any inaccuracies. All the criticism was constructive, which helped to optimize our study. We are resubmitting our MS after major revision, based on the reviewers' comment. Corrections in the main text are indicated in proofreading. Changes in the MS in response to each comment in the answers are also indicated.

We hope the reviewer will find the new version more acceptable from publication.

Reviewer 2:

The manuscript that was submitted for analysis contains some interesting results. The experiments were applied correctly. I would like to express my appreciation for the Authors for a comprehensive approach to research. Various research methods and techniques have been applied, through biochemical methods and to plant physiology. It is very important and definitely facilitates the interpretation. The whole argument is conducted in a coherent and logical way, which makes it easier for the reader to follow the presented results and argue their interpretation.

>>> Thank you very much for your comments.

I have no criticisms of this manuscript. I only ask the Authors to remove their initials from the methodology :) I mean 4.5. ‘Kami’ pigment, please change to Pigments. ‘King’ should be removed 4.6. Sample preparation and GC metabolomics analyzes Kinga

>>> Thank you for your notification, the titles of section 4.5 and 4.6 have been corrected.

Round 2

Reviewer 1 Report

I am satisfied by all authors' corrections and I recommend the manuscript to be accepted for publication.